# In-N-On: Scaling Egocentric Manipulation with in-the-wild and on-task Data

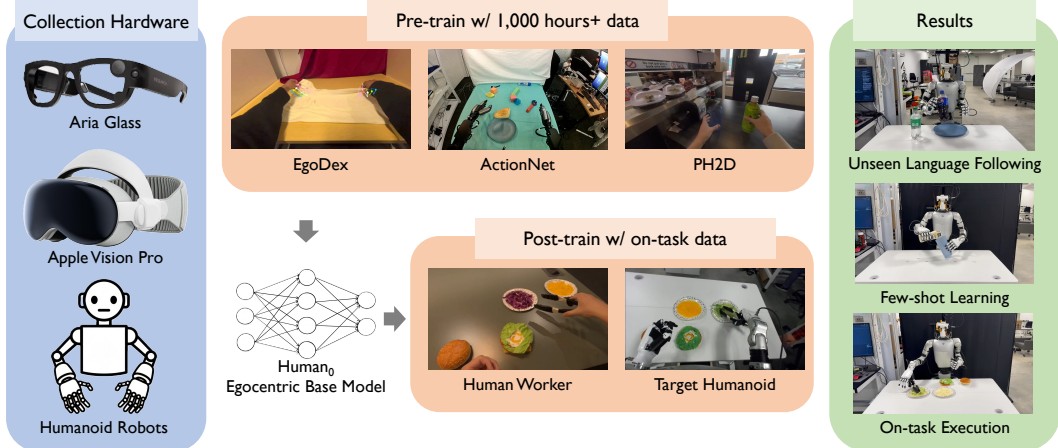

Figure 1: This paper investigates large-scale pre-training and post-training with egocentric human data. We curate a large-scale **P**hysical **H**uman-humanoid**S D**ataset, dubbed $PH^S D$ to train a base model to model egocentric human-humanoid behavior. Empirically, we show that post-training $Human_0$ achieves several interesting properties, including strong language following of instructions unseen in robot data, few-shot execution, and improved on-task performance.

## Abstract

Egocentric videos are a valuable and scalable data source to learn manipulation policies. However, due to significant data heterogeneity, most existing approaches utilize human data for simple pre-training, which does not unlock its full potential. This paper provides a recipe for collecting and using egocentric data by categorizing human data into two categories: **in-the-wild** and **on-task**. We first curate a dataset, $PH^S D$, which contains over 1,000 hours of diverse in-the-wild egocentric data and over 20 hours of on-task data directly aligned to the target manipulation tasks. This enables learning a large egocentric language-conditioned flow matching policy, $Human_0$. We further adopt domain adaptation techniques to align the gap between humans and humanoids. Empirically, we show $Human_0$ achieves several novel properties, including language following of instructions from only human data, few-shot learning, and improved robustness using on-task data. For full reproducibility, we plan to release the dataset, base weights, and code upon acceptance. Anonymized project website: https://human0-anonymous.github.io

## 1 Introduction

The robot manipulation community has recently witnessed great progress in learning from real robot demonstrations (Black et al., 2024; Intelligence et al., 2025; Liu et al., 2024; Barreiros et al., 2025; Kim et al., 2025; Zhao et al., 2024). Behind the curtain are novel algorithms (Intelligence et al., 2025) and large-scale robot data (O'Neill et al., 2024; Black et al., 2024), which enable dexterous and long-horizon tasks (Barreiros et al., 2025). However, existing foundational manipulation polices

still lack *robust real-world generalizability* compared to their counterparts in LLM (Achiam et al., 2023) or self-driving (Tesla, 2025) that are trained on much larger-scale data.

In search of a novel data source to fuel model training, researchers have turned to cross-embodiment learning from different robots (O'Neill et al., 2024; Kim et al., 2024; Black et al., 2024), and, more recently, to human data (Ma et al., 2024; Grauman et al., 2022; 2024; Banerjee et al., 2025). Intuitively, humans are naturally the most prominent physical embodiment compared to other morphologies that can easily manipulate daily objects. Learning from human data has been studied for over a decade. Modular methods learn affordance (Wang et al., 2017; Mendonca et al., 2023; Bahl et al., 2023), plan robot manipulation in a model-based fashion (Li et al., 2024; Xiong et al., 2021; Qin et al., 2022), and planning (Chen et al., 2021). More recently, advances in computer vision have enabled much more precise finger keypoint tracking, recent methods (Kareer et al., 2025; Qiu et al., 2025; Luo et al., 2025; Yang et al., 2025; Niu et al., 2025; Lepert et al., 2025; Qin et al., 2022) have shown that they can be used in an end-to-end manner, which has the potential to be easily scaled up.

However, the vast amount of human data also leads to significant data heterogeneity. Existing human datasets are very diverse - ranging from daily activities such as walking and dancing (Grauman et al., 2022), long-horizon kitchen activities (Damen et al., 2018), and even sitcoms (Wang et al., 2017). To address such heterogeneity (or misalignment between embodiments), some methods propose new algorithms to use intermediate representations such as object pose (Li et al., 2024) or affordance (Bahl et al., 2023) to learn from these **in-the-wild** datasets. On the other hand, recent end-to-end approaches (Yang et al., 2025; Luo et al., 2025; Bi et al., 2025) have resolved to using these datasets only for pre-training. This is a compromise because simple fine-tuning with highly heterogeneous data leads to sub-optimal performance.

Apart from efforts in improving algorithms, recent methods (Qiu et al., 2025; Kareer et al., 2025; Wang et al., 2024; Xu et al., 2025; Tao et al., 2025) have also focused on collecting **on-task** human data. Instead of recording daily activities, on-task data collections require human demonstrators to perform exactly the tasks that robots will be working on. This results in the curation of task-oriented and segmented training demonstrations that are well-aligned to the target deployment distribution. The level of alignment has been empirically shown to be enough for co-training (Qiu et al., 2025; Kareer et al., 2025; Tao et al., 2025; Xu et al., 2025).

We believe that it is important to use both in-the-wild and on-task data to unlock the full potential of human data: in-the-wild data is easy to collect and diverse, but it may be suitable only for bootstraping a base model. In contrast, on-task data is more well-aligned with the target distribution but smaller in magnitude. Consider a scenario, where we want to use a robot to replace a tedious burger-making job at a frozen food processing factory. On-task data can be collected by having the workers wear data collection device, which would give data aligned to the target distribution

This paper investigates the boundary between these two paradigms. Our insight is to use **in**-the-wild data **and on**-task data for pre-training and post-training. With language annotations and a unified human-centric action space (Qiu et al., 2025), this enables learning of a large language-conditioned flow matching policy, $Human_0$. $Human_0$ is dedicated specific for mobile manipulators, which often have access to just egocentric vision without third-person camera. In addition, we also use domain adaptation technique to regularize the training process to improve generalization and alleviate overfitting to a specific domain or camera configuration.

We evaluate $Human_0$ on a real Unitree H1 humanoid and a Unitree G1 humanoid equipped with 5-fingered dexterous hands. Empirically, the pre-training and post-training for $Human_0$ achieve several novel properties, including language following of instructions that are unseen in the robot training data and few-shot learning, which is validated by systematic ablation studies. In particular, we studied a task, *fast food worker*, that potentially has real-world economic values. We show how the on-task data collected for this practical scenario improves policy robustness drastically.

In sum, our contributions are,

- A large-scale human-humanoid dataset, $Human_0$, that provides data recipe for pre-training and post-training an egocentric model. We plan to open-source the dataset.
- A base egocentric manipulation model, $Human_0$, which is augmented with the domain adaptation technique and applicable to many egocentric bimanual embodiments. The weights will be open-sourced.

- Extensive experimental results with demonstrations of language following and few-shot learning on real humanoid robots.

## 2 RELATED WORK

**Large-scale Manipulation Models.** Recent advances in vision-language-action (VLA) models have shown promising progress in robotic manipulation tasks, with a growing emphasis on models' robustness and generalization. Building upon early efforts in learning from real-robot demonstrations (Chi et al., 2023; Zhao et al., 2023), recent methods (Kim et al., 2024; Black et al., 2024; Liu et al., 2024; Intelligence et al., 2025; Barreiros et al., 2025) explored how to scale up robot manipulation policy training with more data. The advances happened both in the modeling regime and the data regime. In the context of modeling, VLAs extend vision-language models (VLMs) or large-language models (LLMs) with action decoders to make use of pre-trained knowledge infused in VLMs. More recently, Intelligence et al. (2025) also proposed a new paradigm to make the training process more data-efficient. On the other hand, data is important for scaling up the manipulation model. Notably, many large manipulation models (Liu et al., 2024; Black et al., 2024) rely on cross-embodiment learning (O'Neill et al., 2024), where a model designs its architecture specifically to work with data from multiple robot embodiments. However, even with cross-embodiment learning, the magnitude of available data is still significantly smaller compared to counterparts in language or vision models. Current manipulation models are data-hungry for more generalizability.

**Learning from Human Videos.** Learning robot policies from human videos has been an active research direction, driven by the availability of large-scale human data. Early efforts (Nair et al., 2022; Radosavovic et al., 2023; Ma et al., 2022) focused on leveraging human videos to pre-train visual representations that are better suited for downstream manipulation policy learning; or to leverage human videos to learn intermediate representations such as affordance (Bahl et al., 2023). Beyond pre-training on visual tasks for improved initializations, other works (Wang et al., 2023; Bharadhwaj et al., 2024a; Xiong et al.; Wen et al., 2023; Bharadhwaj et al., 2024b; Bahl et al., 2023) attempt to use human data directly for downstream tasks such as point tracking (Bharadhwaj et al., 2024b; Wen et al., 2023), and high-level planner (Wang et al., 2023), which are then used to guide robot action prediction.

**End-to-end Learning Manipulation Policies from Human.** An increasing number of works have started to investigate scaling manipulation in an end-to-end manner by leveraging human demonstrations (Kareer et al., 2025; Bi et al., 2025; Qiu et al., 2025; Yuan et al.; Zhu et al., 2025; Tao et al., 2025; Punamiya et al.). They either use diverse **in-the-wild** data for pre-training (Bi et al., 2025; Yang et al., 2025; Luo et al., 2025) or **on-task** data for co-training. Notably, Lepert et al. (2025) modular vision modules to edit human videos to match robot videos to reduce visual gaps. Concurrently, EMMA Zhu et al. (2025) learns a mobile manipulation policy using human data. However, there has yet to be an attempt to explore both in-the-wild data and on-task data to cover both pre-training and post-training stages. This paper aims to bridge such a gap by prescribing a recipe for data curation, an end-to-end large egocentric manipulation base model, and algorithmic advances to improve the model.

## 3 METHOD

This paper discusses models and data recipes for pre-training and post-training a base model for egocentric manipulation. Sec. 3.1 describes the curation process for a large human-humanoid dataset. Sec. 3.2 discusses the design choices for the base model, including unified state-action space and domain adaptation.

### 3.1 PH$^S$D: PHYSICAL HUMANS-HUMANOIDS DATASET

To construct a dataset suitable for learning egocentric policies, we curate and process data from multiple sources into a unified format. We plan to open-source the dataset for full details. Here, we describe the curation procedure concisely and discuss design choices.

**Unified human-centric state-action space.** Different robot datasets have different robot configurations. The kinematics are very different (*e.g.,* different number of joints). While some cross-

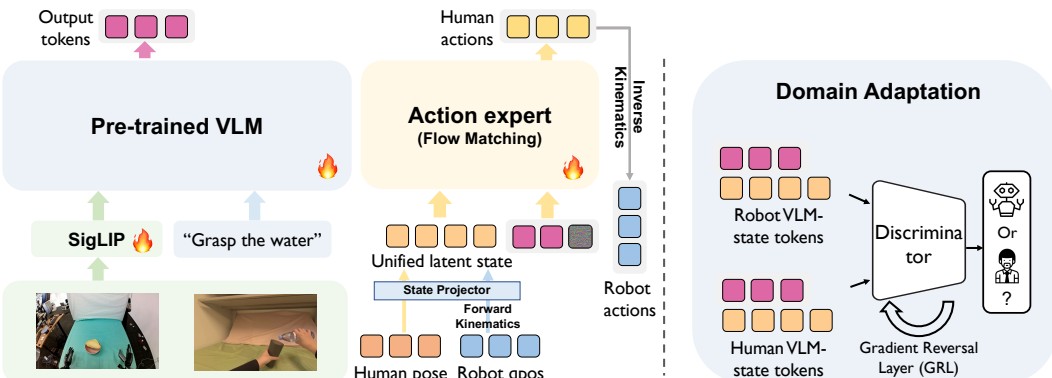

Figure 2: Method overview. Our approach follows a two-stage training recipe: (1) pre-training on large-scale in-the-wild human and robot data that are mapped into a unified human-centric state-action space; and (2) on-task post-training using task-aligned human and robot demonstrations. To bridge the embodiment gap, We employ a domain-adversarial discriminator that takes SigLIP visual features and action-state embeddings as input and predicts whether a sample is from human or robot data. Through gradient reversal, this encourages the policy's encoders to produce embodiment-invariant representations, enabling effective transfer between human and robot observations.

embodiment learning methods Liu et al. (2024) use attention-mask in a long preset vectors, they are not principal as the physical interpretations of different embodiments are not well-aligned. To address this issue, we choose to process the state and action space of all data into the most prevalent embodiment - humans. We use the same convention as human policy (Qiu et al., 2025):

- **Human head pose** (6-dim rotation Zhou et al. (2019)): for robots with active neck, we use forward kinematics (FK) to compute head orientation as a 6D rotation vector, which mimics human necks.

- **Human wrist poses** (18-dim): each of the two wrists is parameterized as a 9-dim vector, which is consisted of a 6D rotation vector and a 3D translation vector w.r.t. the head.

- **Human fingertip 3D keypoints** (30-dim): we use the 3D keypoints of each 5-fingered hands as state/action. During inference time, they are translated to hand motor commands with retargeting via Pinnocchio (Carpentier et al., 2019). Note that this can be trivially mapped to gripper using heuristics such as thumb-index distance. While some other parameterization, such as MANO (Yang et al., 2025), is also possible, we find most of the existing humanoid dataset use low-dof hands (Fourier ActionNet Team, 2025). Hence, it does not require excessive parameterization.

**In-the-wild datasets.** For this project, we use EgoDex (Hoque et al., 2025), Fourier Action-Net (Fourier ActionNet Team, 2025), and PH2D (Qiu et al., 2025) for pre-training.

- EgoDex (Hoque et al., 2025) contains 800+ hours of skill-rich human demonstrations, which were collected using multiple Apple Vision Pros. It contains 6dof head pose, wrist pose, and finger keypoints, which can be easily processed into our human-centric state-action space.

- The ActionNet dataset (Fourier ActionNet Team, 2025) contains over 100 hours of humanoid demonstrations - most of which were done on the Fourier GR1 robot embodiment equipped with bimanual Fourier 5-fingered 6-DoF dexterous hands. To obtain human-centric representation from this dataset, we use forward kinematics to compute SE(3) transformations of the GR1 head pose, wrist pose, and finger keypoints on the training demonstrations.

- The PH2D (Qiu et al., 2025) dataset contains human and humanoid demonstrations of various tasks. Similar to EgoDex (Hoque et al., 2025), PH2D also collected human data with Apple Vision Pro, which can be processed in a similar manner. The humanoid data

(collected on Unitree H1 with 5-fingered Inspire hands) are also processed to the human-centric representation similar to ActionNet.

**Data collection hardware and annotations.** To ensure high-quality hand poses in our on-task datasets, we use commercial-grade data collection devices including Apple Vision Pro and the Meta Aria Glass. For Vision Pro, we use ARKit predictions following existing work (Qiu et al., 2025). For Meta Aria glass, we use the ARIA MPS Service (Engel et al., 2023) to generate wrist poses and dense fingertip keypoint predictions. Since the Aria glass provides camera poses and calibrated intrinsics, the poses can be trivially transformed to our human-centric representation. Visualizations and projections of these datasets can be found in the accompanying website.

### 3.2 HUMAN$_0$: FOUNDATIONAL HUMAN-HUMANOID BASE MODEL

#### 3.2.1 ARCHITECTURE

While the pre-training and post-training recipes proposed in this paper are model-agnostic, we adopt a language-conditioned flow matching model (Black et al., 2024). Specifically, a SigLIP-based vision module extracts visual tokens $v \in \mathbb{R}^{L \times C}$, where $L$ is the number of patches and $C$ is the embedding dimension. The SigLIP encoder provides strong alignment between visual inputs and text, enabling downstream instruction grounding. Visual tokens are then combined with text embeddings $n \in \mathbb{R}^{T \times C}$ to form a joint multi-modal representation, which is further processed in transformer blocks to propagate cross-modal context.

To use the human-centric representation, we use lightweight MLPs to encode input states and the output actions. For the input states, the physically interpretable human-centric state is projected to a pose latent $x \in \mathbb{R}^C$. We denote the latent tokens produced by the backbone transformer as

$$z = \text{Transformers}(v, n, x), \quad z \in \mathbb{R}^C, \tag{1}$$

which integrate information across modalities. Unless otherwise noted, we use the pre-trained checkpoint released by Black et al. (2024) to initialize the model pre-training. Note that since the human-centric representations introduce larger vector sizes and different interpretations of each element at different indices, we swap out the original projection modules with different dimensions and random initialization.

#### 3.2.2 PRE-TRAINING

We first pretrain the base model using over 1,000+ hours of mixed data from EgoDex (Hoque et al., 2025), ActionNet (Fourier ActionNet Team, 2025), and PH2D (Qiu et al., 2025), covering rich egocentric human and robot manipulation scenarios. During this stage, the objective is to learn a unified vision–language–action prior that models human-like behaviors across different embodiments.

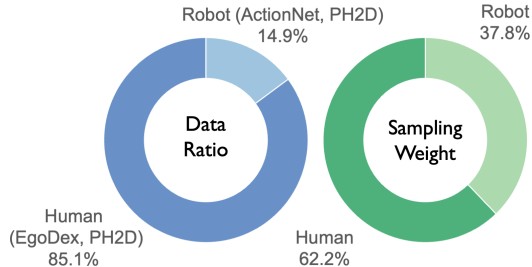

Figure 3: Data size ratio and sampling factor for **pre-training** human-humanoid data.

More concretely, let $a \in \mathbb{R}^C$ be the target action, the model is trained with a flow-matching objective in an end-to-end manner:

$$\mathcal{L}_{\text{FM}} = \mathbb{E}\left[\, \|f_\theta^{\text{flow}}(z, a_t, t) - (a - u)\|_2^2 \,\right], \tag{2}$$

where the Gaussian noise vector $u \sim \mathcal{N}(0, I_C)$, time step $t \sim U(0, 1)$, and interpolated action $a_t = (1 - t)u + ta$. The $f_\theta^{\text{flow}}(z, a_t, t)$ represents the predicted flow vector, which points from the noisy sample towards the target. This pre-training stage equips the base model with broad visuomotor priors from the vast amount of human videos. In addition, it aligns the VLM originally trained on image-text data with human prediction. The shared embodiment space provides a strong regularization for post-training, enabling effective transfer between human demonstrations and robot executions.

**Data mixing recipe.** The distribution of the pre-training data is presented in Fig. 3. Note that due to the overwhelming amount of human data in pre-training, we manually adjust the training data sampler ratio to balance and stablize the training process.

### 3.2.3 POST-TRAINING

During post-training, we focus exclusively on human and robot data collected for the task of interest. The goal is to refine the policy's language grounding and visuomotor control to match the distribution of real-world tasks, where data can be collected by actual human workers performing these real-world tasks. Thanks to our unified action space design, the training procedure follows Eq. equation 2 precisely.

**Data mixing recipe.** The distribution of post-training data is presented in Fig. 4. Compared to pre-training, our post-training dataset has considerably more robot data. Empirically, we found that sampling slightly more often (*e.g.*, 70%) from the human data helps preserve semantics in human data better. This finding is somewhat consistent with Tao et al. (2025), which used an 8:2 sampling ratio to sample human data more often.

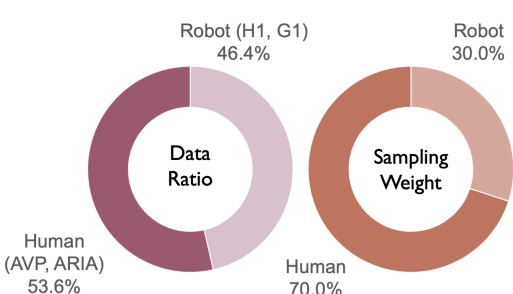

Figure 4: Data size ratio and sampling factor for **post-training** human-humanoid data.

In summary, our pre-training and post-training process enables various interesting properties, including (1) language following of instructions unseen in robot data; (2) few-shot robot data learning with as few as 1 demonstration; and (3) improved robustness across related tasks.

### 3.2.4 DOMAIN ADAPTATION: BLURRING THE LINE BETWEEN EMBODIMENTS

Different human-humanoid embodiments use different cameras and have different kinematic configurations. Although we use image augmentation and human-centric representation with forward kinematics to provide regularization, the model may still learn to distinguish different embodiments, resulting in overfitting to a specific configuration.

To discourage the model from overfitting to specific visual cues or proprioceptive cues, we introduce a discriminator network (Ganin et al., 2016). Specifically, the network is tasked to classify the type of embodiment, and we use the Gradient Reversal Layer (GRL) (Ganin et al., 2016) to discourage successful classification. Hence, the GRL provides additional supervision so that the latent features obtained from the visual backbone and state projection are more embodiment-agnostic.

More specifically, the GRL is trained to differentiate between the feature encoding of human data and those of robot data. We concatenate the visual tokens $v$ from SigLIP encoder with the projected pose latent $x$ along the token dimension, and pass them though a attention head to obtain a feature vector:

$$m = \text{Attn}\big(\text{Concatenate}(v, x)\big), \quad m \in \mathbb{R}^C \tag{3}$$

The feature vector $m$ is then passed through a lightweight discriminator MLP $D_\theta$ that predicts the input's embodiment type. The discriminator is trained with binary cross-entropy loss:

$$\mathcal{L}_D(\phi \mid \theta) = -\mathbb{E}\big[\log D_\phi(m_h)\big] - \mathbb{E}\big[\log(1 - D_\phi(m_r))\big], \tag{4}$$

where $m_h$ and $m_r$ denote feature vectors obtained from human and robot data, respectively. With a GRL inserted between feature vectors and discriminator $D_\phi$, the optimization is adversarial: $D_\phi$ minimizes $\mathcal{L}_D$, while the backbone policy encoders $f_\theta$ maximizes it:

$$\min_\theta \max_\phi \mathcal{L}_D(\phi, \theta) \tag{5}$$

In other words, the GRL encouraging the upstream policy encoder to produce features that are invariant to the human-robot domain distinction. This adversarial setup promotes feature alignment across data domains and embodiments, enabling more effective transfer of manipulation behaviors between human demonstrations and robot.

**Final Loss.** Combining both flow matching L2 loss and the domain adaptation loss, the final training loss is given by

$$\mathcal{L}_{\text{final}} = \mathcal{L}_{\text{FM}} + \lambda \cdot \mathcal{L}_D(\phi \mid \theta) \,, \tag{6}$$

where $\lambda$ is a hyperparameter balancing the scale of flow matching loss and discriminator loss.

## 4 EXPERIMENTS

### 4.1 EXPERIMENTAL SETUP

**Implementation Details.** For raw human-humanoid data, we use timestamps to synchronize episodes. We process the states and actions into the human-centric representation, and reduce the width of the images to up to 320 for efficient storage and data loading. To obtain the base $\text{Human}_0$ model, we train on 8 H200 GPUs for 100k steps using 160 batch size. The weights are initialized with pre-trained checkpoint (Black et al., 2024). As for post-training, we fine-tune the trained base model on a single H100 GPU for 30k steps using 10 batch size. This demonstrates one potential application of our base model to democratize egocentric manipulation training with just a single GPU.

**Robot Platforms.** For data collection and policy deployment, we use a Unitree H1 and a Unitree G1 humanoid robot. Both robots are equipped with Inspire 5-fingered dexterous hands. We use OpenTV Cheng et al. (2024) as the teleoperation software.

**Baselines.** We compare with 4 baseline models. $\pi_0$ (Black et al., 2024) is a language-conditioned flow matching model trained on many robot embodiments, which is also the initialization we use before pre-training. GR00T N1 (Bjorck et al., 2025) is another language-conditioned VLA using diffusion transformers. HAT (Qiu et al., 2025) trains specialist policies and is thus unsuitable for pre-training or tasks that require language conditioning. Finally, $\text{Human}_0$ w/o human follows the same pre-training and post-training procedure, but without any human data.

**Experimental Protocol.** We experiment with 4 different humanoid manipulation tasks with *in-distribution (I.D.)* and Out-Of-Distribution (O.O.D.) settings. The I.D. setting tests the learned skills with language, scenes, and objects that approximately resemble corresponding sequences in the robot training demonstrations. The O.O.D. setting tests configurations that are unseen in the robot training data, but may present in human data.

The tasks are illustrated in Fig. 5. Objects used in these tasks are visualized in Sec. A.1. Each task is designed to serve a different evaluation purpose. Specifically,

- **Single object grasping** is a sanity check task. The robot is placed in front of a table with an object and a container. The robot is tasked to pick up the object, and place it into the container.

- **Multi object grasping** is an extension of the single object grasping, where we add distractor objects. As shown in Fig. 5, the robot is tasked to grasp the object specified by the language instruction. One challenge here is that the distractors are also presented in the training data. Hence, this task evaluates the ability of the robot to perform language following.

- **Burger assembly** is intended to mimic a real-world task, where a worker at a fast food restaurant or at a food processing facility assembles a burger based on language instructions. The task is long-horizon, which involves multiple steps from using tongs to pick up ingredients specified by language, and putting the top bread. In addition, collecting on-task human data for this application can be hypothetically done by having the actual workers use wearable devices.

- **Pouring** shows the few-shot learning capability of our model. Compared to previous tasks that have hundreds of robot sequences per task, we use only **1** robot training demonstration in the bimanual pouring task, to demonstrate how human demonstrations enable few-shot robot learning.

| Method | Single Object Grasping | | Multi Object Grasping | | Burger assembly | | Pouring |
|---|---|---|---|---|---|---|---|
| | I.D. | O.O.D | I.D. | O.O.D | I.D. | O.O.D | I.D. |
| $\pi_0$ | 19/20 | 19/20 | 25/30 | 16/30 | 5/12 | 3/12 | 0/20 |
| GR00T N1 | 18/20 | 13/20 | 6/30 | 8/30 | 4/12 | 3/12 | 0/20 |
| HAT w/ human | 17/20 | 15/20 | - | - | - | - | 2/20 |
| $\text{Human}_0$ w/o human | 18/20 | 18/20 | 23/30 | 15/30 | 7/12 | 2/12 | 2/20 |
| $\text{Human}_0$ (Ours) | **20/20** | 19/20 | **29/30** | **30/30** | **8/12** | **7/12** | **5/20** |

Table 1: Baseline comparison results. Our method achieves the best performance among all baselines across the four manipulation tasks, under both I.D. and O.O.D. settings. We also show that training with large-scale human data improves model performance.

| Discriminator | Staged Pouring SR | | | |
|---|---|---|---|---|
| | Right grasp | Left grasp | Pour | SR |
| ✗ | **17/20** | 5/20 | 3/20 | 15% |
| ✓ | 16/20 | **7/20** | **5/20** | **25%** |

Table 2: Ablation study of domain adaptation using the pouring task. Pouring task is a challenging bimanual task that can be divided into 3 stages. The success rates (SR) reported are compositional.

## 4.2 EVALUATION

### 4.2.1 MAIN EXPERIMENT

**Flow matching performs slightly better on single object grasping.** As shown in Tab. 1, all policies achieve similar results in the I.D. setting for the single object grasping task. However, in the more challenging O.O.D. setting, flow-matching-based policies, including $\pi_0$ (Black et al., 2024) and ours, outperform other methods.

**Pre-training and post-training on mixing human-humanoid data improves language following drastically.** In the more challenging multi-object grasping task, where the robot needs to grasp an object with distractors present, $\text{Human}_0$ performs much better than other baselines. Surprisingly, even in the O.O.D. case, where the robot is operating in unseen scenes or objects that are presented only in human data, $\text{Human}_0$ still achieves great success rate, with emerging behavior such as tracking and re-trying. A detailed ablation on different backgrounds is also presented in Sec. A.2.

**Post-training with on-task data improves performance on challenging task.** In the difficult burger assembly task, where the robot needs to perform multiple steps with tool usage, $\text{Human}_0$ outperforms baseline methods with over 100% relative improvement. Notably, in the O.O.D. scenarios, we intentionally task the robot to manipulate ingredients unseen in the robot data (*i.e.,* red cabbage and swiss cheese) to mimic special requests to food workers in the real world.

**$\text{Human}_0$ enables 1-shot robot data learning.** With just a single robot demonstration of the bimanual pouring task, $\text{Human}_0$ achieves 5/20 success rate. Note that the model has never seen a Unitree G1 performing this bimanual pouring behavior in either pre-training or the post-training stage. While the model succeeds occasionally, few-shot or even zero-shot learning remains a challenging task. One exciting direction is to further scale the pre-training and post-training stage to continue pushing the success rate of few-shot learning.

### 4.2.2 ABALATION STUDY

**Human data helps.** From Tab. 1, we can see that when $\text{Human}_0$ is trained without human data and only on humanoid data, the performance degrades significantly.

**Domain Discriminator reduces the embodiment gap and helps few-shot learning.** Tab. 3 shows how the domain discriminator helps improve the few-shot learning case drastically. Without a domain discriminator, the model quickly overfits to the single humanoid demonstration over the training trajectory.

**Performance dynamics with robot data as a variable.** Fig. 6 shows the performance change with varying number of robot data used in training. This setting is intended to show how human data

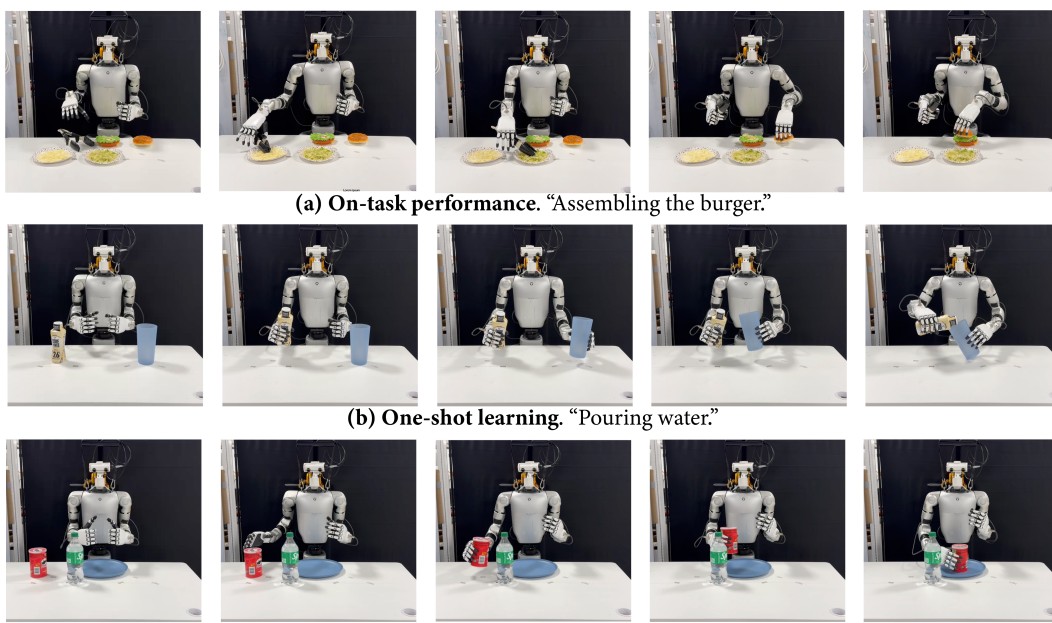

(a) **On-task performance**. "Assembling the burger."

(b) **One-shot learning**. "Pouring water."

(c) **Following language instruction available only in human data**. "Grasping the Pringles can."

Figure 5: We task the robot to perform several manipulation tasks to test its ability to perform few-shot learning, language instruction following, and robustness using on-task human data. Videos are available at website. (Top to bottom: burger assembly, pouring, and multi-object grasping).

| Discriminator | Staged Pouring SR | | | |
|---|---|---|---|---|
| | Right grasp | Left grasp | Pour | SR |
| ✗ | **17/20** | 5/20 | 3/20 | 15% |
| ✓ | 16/20 | **7/20** | **5/20** | **25%** |

Table 3: Ablation study of domain adaptation using the pouring task. Pouring task is a challenging bimanual task that can be divided into 3 stages. The success rates (SR) reported are compositional.

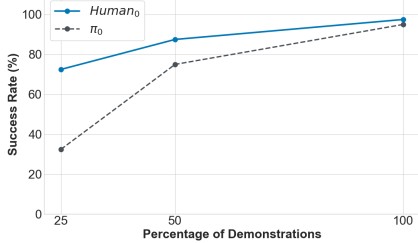

Figure 6: **Data efficiency.** Performance on single object grasping. The x-axis represents the percentage of available single-object grasping robot demonstrations used in training.

can help even in simple tasks like single-object grasping. Robot hardware evolves all the time, but human data will always be useful.

**Qualitative results.** For more qualitative results such as videos, we encourage the readers to check out the accompanying website provided at the end of the abstract.

## 5 CONCLUSION

In this work, we presented IN-N-ON, a scalable recipe for leveraging egocentric human data through a principled taxonomy that distinguishes in-the-wild and on-task data. With $PH^SD$, a large-scale dataset comprising over 1,000 hours of diverse in-the-wild human and humanoid demonstrations and 20+ hours of task-aligned data, we enabled the training of $Human_0$. $Human_0$ demonstrates several novel properties of scaling human data with language annotations. There are several interesting future directions, including further scaling of human data and testing on different robot embodiments other than humanoid robots.

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

# A APPENDIX

## A.1 OBJECT VISUALIZATION

We visualized the objects used in each task in Figs 7, 8, 9 and 10.

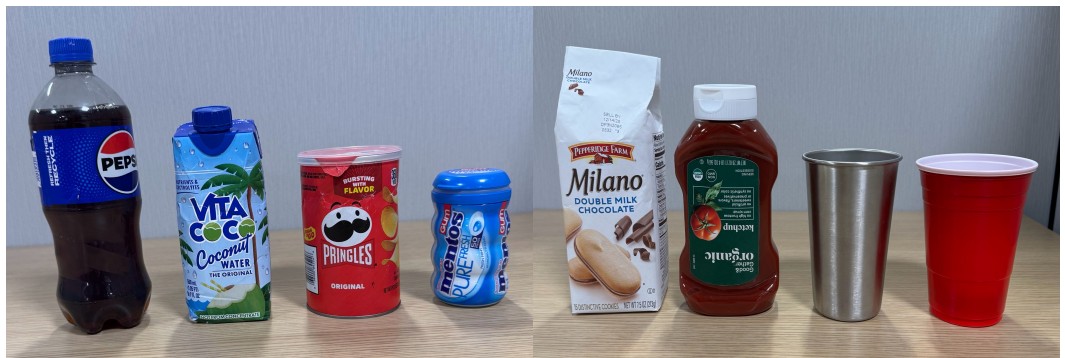

Figure 7: Single Object Grasping: 4 seen objects and 4 unseen objects

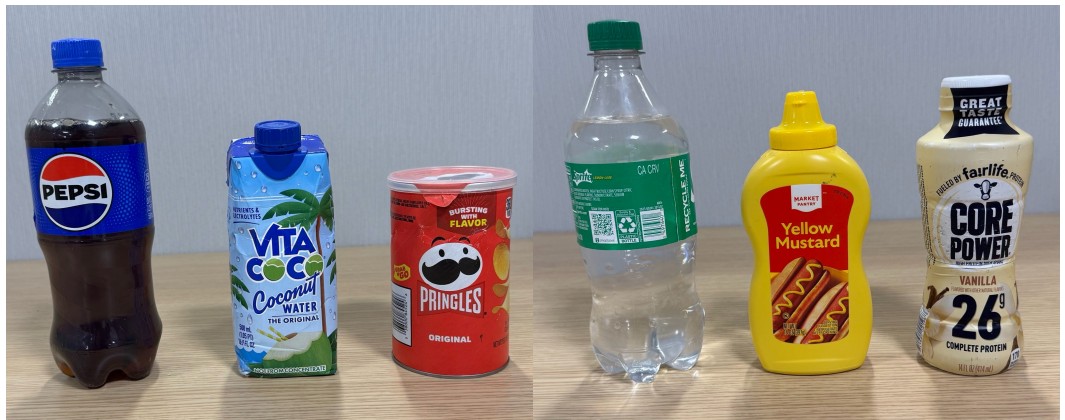

Figure 8: Multi Object Grasping: 3 seen objects and 3 unseen objects. The unseen objects appear in the human data with corresponding instructions.

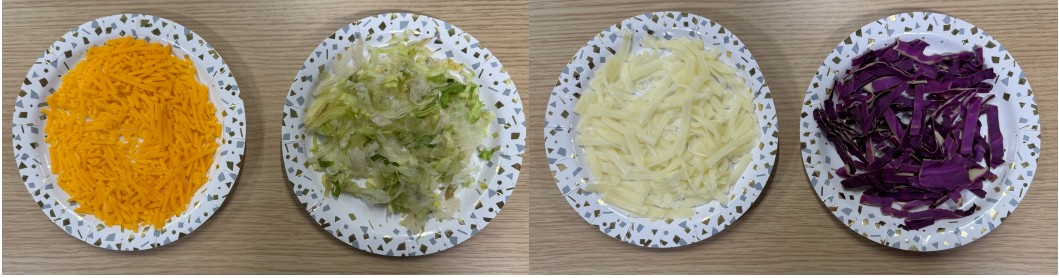

Figure 9: Burger Assembly: 2 seen objects and 2 unseen objects. The unseen objects appear in the human data with corresponding instructions.

## A.2 BACKGROUND ABLATION

With scaled pre-training and post-training, Human$_0$ achieves impressive background generalization capability. In Tab. 4, we test multi-object grasping under different backgrounds.

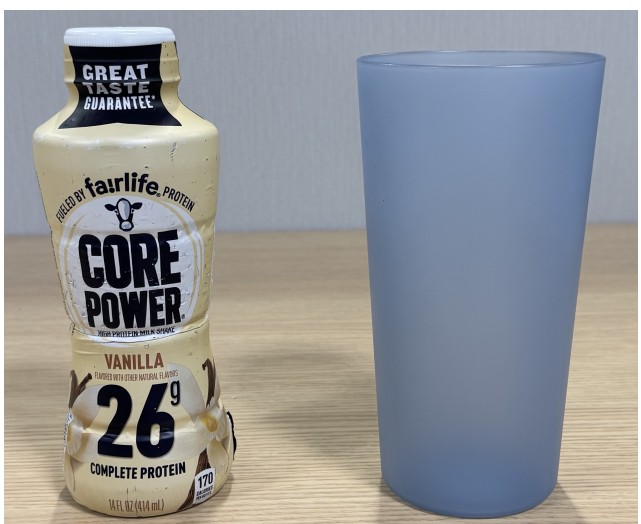

Figure 10: Pouring: protein drink and plastic cup

| Background | Multi Object Grasping | |
|---|---|---|
| | I.D. | O.O.D |
| White table (original) | 29/30 | 30/30 |
| Black tablecloth | 26/30 | 29/30 |
| floral tablecloth | 24/30 | 28/30 |

Table 4: Ablation study of manipulation performance under different background.

### A.3 MULTI-TARGET LANGUAGE FOLLOWING

Here we test the language following ability of $Human_0$ under both I.D. and O.O.D settings in Tab. 5.

| Method | Multi Object Grasping | | Burger assembly | |
|---|---|---|---|---|
| | I.D. | O.O.D | I.D. | O.O.D |
| $Human_0$ (Ours) | 29/30 | 30/30 | 10/12 | 10/12 |

Table 5: Evaluation of language following

## B USE OF LLM

The authors used large language models (LLMs) solely for minor wording refinements during the writing process. All scientific content, technical ideas, experiment results, and original writing were produced by the authors.

