# OpenReview forum: "In-and-On: Scaling Egocentric Manipulation with in-the-wild and on-task Data"
_ICLR.cc/2026/Conference — ICLR 2026 Conference Withdrawn Submission_

### Official Review · Reviewer_RBY4 · 2025-10-19

**Soundness:** 2
**Presentation:** 2
**Contribution:** 1
**Rating:** 2
**Confidence:** 5

**Summary:**

This paper introduces a cross-embodiment humanoid policy, Human$_0$. To address the embodiment gap between humans and robots, the authors propose a two-stage approach. In the first stage, they use in-the-wild human and robot datasets to build an ego-centric base model. After that, they post-train their policy on task-specific data using both robot and collected human demonstrations. During training, they adopt a discriminative module to make the model more robust to the human-robot domain gap. For cross-embodiment training, they also provide a curated training dataset, PH$^S$D. Their method outperforms other baselines even in few-shot learning scenarios.

**Strengths:**

- **Provide cross-embodiment dataset**
  - This paper provide PH$^S$D, a curated and processed dataset for embodiment-agnostic policy learning. Using public human and robot datasets, the authors construct a unified labeling scheme between human and robots. This contribution could promote progress in the cross-embodiment robot learning community.
- **Simulating real-world scenario**
  - To simulate real-world applications, the authors design a task called *Burger Assembly*, assuming that employees wear data-collection devices to record human data. This task design effectively demonstrates a practical use case for real-world setting.

**Weaknesses:**

**Major Weaknesses**
- **Insufficient comparison with previous works**
  - Human$_0$ adopts a two-stage training process divided by dataset type: in-the-wild data and task-specific data. Beyond this, Human$_0$ is less representative compared to previous work such as EgoVLA[1] and HAT[2]. EgoVLA used human data for pre-training, while HAT used human data for co-training. Both also adopted a unified action space to leverage human data during training. Therefore, the proposed pre-training-then-post-training approach with a unified action space is less persuasive as a contribution of the paper.
- **Insufficient analysis on domain adaptation**
  - In the paper, the authors introduces discriminator module to enhance robustness on different embodiments. They argue that the ablation study in Tab. 3 supports the benefit of this domain adaptation approach. However, based on the result, using discriminator only increases the number of successful trials by two. which is not drastic improvement. It is hard to say that, without using discriminator, the policy is overfitted to a single training trajectory. In fact, if it were overfitted to a single trajectory, it might even perform better in the I.D. setting.
  - The paper also provides limited analysis of the discriminator's design. First, the investigation of using GRL is missing. As the discrimination module, simpler alternatives such as an MLP could have been explored. However, the authors neither justify their choice of GRL nor include an ablation study for it. Moreover, since the domain adaptation loss is a binary classification loss that distinguishes between human and robot data, it cannot be applied to the *Human$_0$ w/o human* baseline.
---
**Minor Weaknesses**
- **Paper organization**
  - Overall, the paper is not well organized.
    - Tab.2 and Tab.3 are duplicated. which critically hinders readability.
    - In L104, Human$_0$ might actually refer to PH$^S$D.
    - There are several typos and irregular spaces throughout the paper.
- **Missing experimental details**
  - Some experimental details are missing. Several training hyperparameters are not described, and the deployment settings lack clarity. Specifically, it is not elaborated which robot is used for each task. For example, in the pouring task, L418 mentions that the Unitree G1 is not used for either pre- or post-training, but this information is not explicitly stated elsewhere in the paper.
---
[1] Yang, Ruihan, et al. "Egovla: Learning vision-language-action models from egocentric human videos." arXiv preprint arXiv:2507.12440 (2025).

[2] Qiu, Ri-Zhao, et al. "Humanoid Policy~ Human Policy." arXiv preprint arXiv:2503.13441 (2025).

**Questions:**

- How does the performance change if the discrimination module is replaced with alternative designs?
- What value of $\lambda$ was used during training?
- How was *Human$_0$ w/o human* trained?
- Please include further comparison with previous works.

---

### Official Review · Reviewer_4ih9 · 2025-10-25

**Soundness:** 3
**Presentation:** 3
**Contribution:** 3
**Rating:** 6
**Confidence:** 3

**Summary:**

This paper proposes a method to train robot manipulation planning via pre-training with three existing datasets (EgoDex, ActionNet, and PH2D) followed by post-training with 20 hours of self-collected on-task data performed by humans and target humanoids. All video data are egocentric, and the human data is used in a domain adaptation manner; a gradient reversal layer aligns the distribution of action latent features to minimize the embodiment gap between humanoid and human. The experimental design includes single-object grasping (without language instruction), multi-object grasping, burger assembly (a short sequence of pick and place tasks), and pouring (a challenging bimanual task). Each task was attempted 20 times with each model, demonstrating the superiority of the proposed method against $\pi_0$.

**Strengths:**

### 1. Clear Motivation.
A core problem in recent Vision-Language-Action (VLA) models is the shortage of training data. Egocentric-view human demonstrations are mainly used to solve this issue.

### 2. A steady community contribution.
The paper provides pre-trained and post-trained models with their datasets.

### 3. Outperforming $\pi_0$ in Grasping Tasks and the Pouring Task.
While not applied to complex cloth folding tasks, the method surpasses $\pi_0$'s performance in basic tasks. Despite not tackling complex cloth folding tasks, it exceeds $\pi_0$ in simpler tasks.

**Weaknesses:**

### 1. Tasks are too simple
Tasks are simpler than those demonstrated with $\pi_0$ in the original paper. Finger motion here is not as complex as the shirt-folding task (Black et al., 2024). Results on the shirt-folding task would better illustrate the effectiveness of the gradient reversal layer.

### 2. The alignment of target distribution
The gradient reversal layer is a simple domain adaptation technique with known limitations, such as assuming identical label distributions for human and humanoid datasets. The authors should clarify if the on-task dataset satisfies this condition.

### 3. The task's practicality is overclaimed
The author claims "we studied a task, fast food worker, that potentially has real-world economic values." (Line 99) However, the experimental setup limits the robot to simple tasks, contradicting the claim of studying tasks with real-world economic value.

### 4. Poor error analysis
The experimental results do not include a detailed error analysis, obscuring the method's limitations. Please provide a more detailed quantitative and qualitative evaluation and discussion for error cases.

**Questions:**

1. Is the method work well with shirt-folding? If not, why?
2. Please provide a more detailed statistical information including label distribution of each domain.
3. Please provide a more detailed error analysis if available.

This reviewer will re-evaluate the work after receiving the author response.

---

### Official Review · Reviewer_hiaJ · 2025-10-28

**Soundness:** 3
**Presentation:** 2
**Contribution:** 2
**Rating:** 4
**Confidence:** 5

**Summary:**

In-N-On presents egocentric manipulation results on a humanoid platform. First, they combine existing egocentric datasets (EgoDex, PH2D) and robot datasets (PH2D, ActionNet) and pretrain with a shared action space (head pose, wrist pose, 3D keypoints). Then, the model is post-trained with in-domain robot and human data. Additionally, the model optimizes a discriminator loss for predicting a human or robot sample. The architecture and types of data collected are not particularly novel (model is language-conditioned VLA; datasets are compiled or collected with existing interfaces). The authors show results on 4 tasks, including simpler grasping tasks, a more dexterous pour task, and a longer horizon burger test. Results show that Human0 benefits from human data and performs better for ID and OOD.

**Strengths:**

1. The recipe seems sensible and scalable. It leverages scalable data sources for robot learning.
2. The related work is easy to read, and I believe the authors have sufficiently cited the prior works.

**Weaknesses:**

1. This paper has limited novelty.
2. The Experiments section is a bit short and feels like it is missing many details.
3. Further discussion of ID/OOD for the experiments section would also be useful.
4. Language conditioning is a strong claim, given the limited experiments. Given many different objects, how well does the policy actually follow the language instruction? I would expect more experiments to back up this claim.
5. The experiments don’t really show that the discriminator consistently helps results. The gains seem somewhat minimal, and it’s also hard to tell given the limited evaluations.

**Questions:**

1. Are the baselines post-trained with the same amount of data?
2. How many teleoperated and human demonstrations are collected per each task?
3. Is the model post-trained once, and evaluated for all the tasks? Or are there separate models per task?

---

### Note · Authors · 2025-11-14

I have read and agree with the venue's withdrawal policy on behalf of myself and my co-authors.